# Dynamic Correlation Adjacency-Matrix-Based Graph Neural Networks for Traffic Flow Prediction

**DOI:** 10.3390/s23062897

**Published:** 2023-03-07

**Authors:** Junhua Gu, Zhihao Jia, Taotao Cai, Xiangyu Song, Adnan Mahmood

**Affiliations:** 1School of Artificial Intelligence, Hebei University of Technology, Tianjin 300000, China; 2School of Mathematics, Physics and Computing, University of Southern Queensland, Toowoomba 4350, Australia; 3School of Software and Electrical Engineering, Swinburne University of Technology, Melbourne 3122, Australia; 4School of Computing, Macquarie University, Sydney 2109, Australia

**Keywords:** graph neural networks, dynamic adjacency matrix, multivariate time series, traffic prediction

## Abstract

Modeling complex spatial and temporal dependencies in multivariate time series data is crucial for traffic forecasting. Graph convolutional networks have proved to be effective in predicting multivariate time series. Although a predefined graph structure can help the model converge to good results quickly, it also limits the further improvement of the model due to its stationary state. In addition, current methods may not converge on some datasets due to the graph structure of these datasets being difficult to learn. Motivated by this, we propose a novel model named Dynamic Correlation Graph Convolutional Network (DCGCN) in this paper. The model can construct adjacency matrices from input data using a correlation coefficient; thus, dynamic correlation graph convolution is used for capturing spatial dependencies. Meanwhile, gated temporal convolution is used for modeling temporal dependencies. Finally, we performed extensive experiments to evaluate the performance of our proposed method against ten existing well-recognized baseline methods using two original and four public datasets.

## 1. Introduction

The transportation system is one of the most critical infrastructures in modern cities, supporting the daily life of cities for millions of people to commute and travel. With rapid urbanization and population growth, the transportation system has become more complex. It includes road vehicles, rail transit, and various modes of shared travel (i.e., *bike sharing*) that have emerged in recent years.

However, urban expansion also faces many corresponding problems, such as air pollution, traffic congestion, etc. Early intervention based on traffic prediction is considered to be key to improving the transportation system’s efficiency and alleviating the above problems. With the development of smart cities and smart transportation systems, sensors installed on roads (*with regard to loop detectors*) can sense traffic conditions and record transactions on subway and bus systems, deriving from traffic surveillance videos and even smartphones equipped with GPS receivers. Traffic forecasts are based on these historical traffic data, as well as external factors that influence traffic conditions, such as weather and holidays.

**Example 1.** 
*As shown in Figure 1a, the road network in the map can be considered as a graph. Sensors are nodes with time series of their states as shown in Figure 1b, and the road segments are edges. Therefore, the datasets of the sensors can be thought of as multivariate time series that have long time spans. These kinds of data can easily print a heat map that shows the features of the road segments. Current navigation applications are able to show a real-time heat map, but, using the prediction of multivariate time series, we can create the heat map of the next hour. The feature can be the flow, speed, or travel time that helps the driver to choose the faster route. It can also be an advanced and composite index such as the passenger car unit (PCU) and saturation for management control.*


There are many practical application scenarios that can be studied in the field of traffic prediction, such as the travel time [1,2,3], speed [4,5,6,7,8], congestion [9,10], and traffic flow, which can be divided into taxi flow [11,12,13,14], bicycle flow [15,16], and highway flow [17,18,19,20,21], which are all studied in this paper.

The traditional way to predict time series uses the whole datasets as the input, which is slower and difficult to converge. Deep learning models such as graph convolutional networks use the historical data to train themselves, and then use fewer data as the input for the prediction. For example, if we train a model using 1 month of data, the model is able to predict the next hour using only 1 hour of data. Once the training is carried out, the model can be both faster and more accurate than the traditional method. Graph convolutional networks have proved to be effective in predicting multivariate time series. In addition to physical-level traffic networks, other latent graph structures can be found to improve the prediction of multivariate time series that do not contain an obvious graph structure. Graph structures that come with datasets are not easy to build, and are probably not the most helpful way to express the graph. The DCRNN [22] uses diffusion convolution on the graph to extract spatial dependencies and GRU to extract temporal dependencies, and it also uses PeMS-BAY and METR-LA datasets for the first time. STGCN [23] uses spectral graph convolution and gated one-dimension convolution to extract spatial and temporal dependencies, respectively. They provide fundamental methods for this area in the early stages.

Graph convolution requires an adjacency matrix, which is predefined in the datasets. However, the predefined graph structure has its limitations: it makes the process of information transfer on the graph fixed, but the spatial dependence is dynamic and changes with time. Therefore, some methods, such as the attention mechanism, are introduced to capture the dynamic dependence. The ASTGCN [24] adds spatial and temporal attention to capture dynamics based on former studies. Graph Wavenet [25] constructs an adaptive adjacency matrix through node embedding to assist the original matrix in the graph convolution. Furthermore, the MTGNN [26] and AGCRN [27] abandon the original graph structure and introduce a graph learning module to generate an adjacency matrix with random node embedding.

When the learning environment is harsh, such as an extensive range of data and insignificant node features, the above methods may face failure. Thus, we introduce a new way to construct dynamic adjacency matrices and put them in the encoder–decoder structure. The main contributions of this paper can be summarized as follows:We propose a novel method for constructing an adjacency matrix using a correlation coefficient for graph convolution.We constructed the adjacency matrix using input data accordingly and used a dynamic correlation convolution module to capture spatial and temporal dependencies with an improved GCN and TCN.We made new datasets based on raw traffic data, and experimental results on both original and public datasets show that our model outperforms the baseline methods.

**Organization.** The remainder of this paper is organized as follows. First, we discuss the related works in Section 2. Then, we present the preliminaries and formally define our research problem in Section 3. We further propose the DCGCN model for traffic flow prediction in Section 4. After that, the experimental evaluation and results are reported in Section 5. Finally, we conclude this work in Section 6.

## 2. Related Work

### 2.1. Multivariate Time Series Prediction

After years of effort, the research on traffic prediction has made great progress. According to the processing, these methods can be roughly divided into two categories: classical methods and deep-learning-based methods. Classical methods include statistical methods and traditional machine learning methods. The statistical method is to build a data-driven statistical model for prediction. The most representative algorithms are historical average (HA), auto-regressive comprehensive moving average ARIMA [28], and vector auto-regression [29]. However, these methods require data to satisfy certain assumptions, and time-varying traffic data are too complex to satisfy these assumptions. In addition, these methods are only applicable to relatively small datasets. For traffic prediction, some traditional machine learning methods are proposed. These methods have the ability to process high-dimensional data and capture complex nonlinear relations.

The deep0learning-based approaches studies how to learn a layer model and transform the original input directly into the expected output. Typically, deep learning models stack up basic learnable blocks, or layers, to form a deep architecture that trains the entire network end-to-end. Several architectures have been developed to handle large and complex spatial–temporal data. Generally, a convolutional neural network (CNN) is used to extract the spatial correlation of grid structure data described in images or videos. Graph convolutional networks (GCNs) extend the convolution operation to more general graph structure data, which are more suitable for representing the traffic network structure. In addition, recurrent neural networks (RNNs) and their variants, LSTM [30] or GRU [31], are often used to model temporal correlation.

### 2.2. Graph Adjacency Matrix for Traffic Forecasting

In former studies, the most common way to construct a predefined adjacency matrix needs the distance between each pair of nodes using a threshold Gaussian kernel [32]:(1)$$A_{ij}=\left\{e^{-\frac{d_{ij}^{2}}{\sigma^{2}}},i\neq j\;and\;e^{-\frac{d_{ij}^{2}}{\sigma^{2}}}\geq \alpha ,0,\;\;otherwise,\right.$$
where $d_{ij}$ is the geometric distance or distance on the road network between node *i* and *j*, σ is the standard deviation of distances, and α is the threshold that is used to control the sparsity of the matrix.

Although a predefined graph structure can help the model converge to a good result quickly, it also limits the further improvement of the model due to its stationary state, which has two meanings: the existence of edges is fixed, as well as the weights of the current edges. To address this problem, many methods have been brought up in former studies that are able to construct an adjacency matrix in the model. These methods can be divided into two parts, partly independent methods and completely independent methods, according to whether they need the predefined existences of edges. Partly independent methods include those brought in the ASTGCN, STSGCN [33], and STFGNN [34], which preserve the existences of edges and learn the weights with a masked matrix or attention mechanism. Using a learnable matrix $W\in R^{N\times N}$ is an obvious choice for completely independent methods, but it is extremely difficult for a random matrix to converge to an adjacency matrix that shows the graph structure correctly, so various methods that use node embedding to construct the adjacency matrix have been brought up in AGCRN, Graph Wavenet [25] and MTGNN, as shown in Table 1.

Using node embedding can decrease the number of learning parameters, thus lowering the difficulty. It also uses a threshold to control the sparsity of the adjacency matrix. These methods are able to learn the existences of edges, as well as the weight. Although a certain effect is made, an assumption is also made, which is that the graph structures of all samples are the same. The physical meaning of the differences between the samples is that the beginning time of the time series is different, which means that the above assumption can be interpreted as suggesting that the graph structure of multivariate time series is not time-variant. This goes against well-known facts. For example, peak or off-peak hours, day or night, sunny or rainy day, and workday or weekend and holidays are all time-variant external factors that affect the features in the datasets of traffic domain; thus, it is probably a better method for generating a dynamic adjacency matrix. Several methods that can generate the adjacency matrix by multiplying input data with a smaller matrix in the middle are brought up in the MTGNN and SLCNN [35], as shown in Table 1, where σ is one of the more commonly used activation functions, such as ReLU, tanh, sigmoid, or their combinations, and hyper-parameter α is used as a threshold to control the sparsity of the matrix. *W*s are learnable parameters, $W_{1},W_{2}\in R^{N\times K}$ in the AGCRN, GWN, and MTGNN, and $W_{1},W_{2}\in R^{K\times K}$ in the SLCNN and MTGNN, where K≪N is the preset dimension.

When the learning environment is harsh, such as when there is a large range of data and insignificant node features, the above methods that generate the adjacency matrix by learning the node feature with node embedding may face failure. In fact, the adjacency matrix is just used to present spatial connections among nodes. Besides the given geological information and learning from ground zero, we can also use a statistic method to measure the correlation among nodes.

## 3. Preliminary

In this section, we introduce the statistical method that we used to construct an adjacency matrix for graph convolution, and the problem definition of multi-time-series forecasting using graph neural networks.

### 3.1. Correlation Matrix in Multiple Regression Analysis

In statistics, the Pearson correlation coefficient *r* can be used to measure the similarity between two variables $X,Y\in R^{N}$, where its value is between −1 to 1. The two variables are positively correlated when r>0, and negatively correlated when r<0. The closer |r| gets to 1, the stronger the correlation. Normally, it can be divided into three parts: the variables have low correlation when |r|<0.4, significant correlation when 0.4≤|r|<0.7, and high correlation when 0.7≤|r|<1.

Note that
(2)$$L_{xy}=\sum_{i=1}^{n}\left(x_{i}-\bar{x}\right)\left(y_{i}-\bar{y}\right).$$

Then, correlation coefficient *r* can be defined as
(3)$$r=\frac{L_{xy}}{\sqrt{L_{xx}}\sqrt{L_{yy}}}=\frac{\sum_{i=1}^{n}\left(x_{i}-\bar{x}\right)\left(y_{i}-\bar{y}\right)}{\sqrt{\sum_{i=1}^{n}{\left(x_{i}-\bar{x}\right)}^{2}}\sqrt{\sum_{i=1}^{n}{\left(y_{i}-\bar{y}\right)}^{2}}}.$$

When extended from two to multiple variables, the correlation coefficient between each pair of variables $X_{i},X_{j}\in R^{N}$ can be formulated as a correlation matrix. Firstly, it can be defined as
(4)$$r_{ij}=\frac{L_{x_{i}x_{j}}}{\sqrt{L_{x_{i}x_{i}}}\sqrt{L_{x_{j}x_{j}}}}=\frac{\sum_{k=1}^{n}\left(x_{ik}-\bar{x_{i}}\right)\left(x_{jk}-\bar{x_{j}}\right)}{\sqrt{\sum_{k=1}^{n}{\left(x_{ik}-\bar{x_{i}}\right)}^{2}}\sqrt{\sum_{k=1}^{n}{\left(x_{jk}-\bar{x_{j}}\right)}^{2}}}.$$

Apparently, $|r_{ij}|<1$. Then, the correlation matrix can be defined as
(5)$$R={\left(r_{ij}\right)}_{N\times N}=\left[\begin{array}{cccc} r_{11} & r_{12} & \cdots & r_{1N}\\ r_{21} & r_{22} & \cdots & r_{2N}\\ \vdots & \vdots & \ddots & \vdots \\ r_{N1} & r_{N2} & \cdots & r_{NN} \end{array}\right]$$

### 3.2. Problem Definition

If the variable changes through time, its value can be formulated as a time series. This variable can be one of the useful features in the traffic domain, such as the traffic flow of specific road segments, the average speed of moving vehicles, the occupancy of the road segment, etc. Each monitor or sensor is usually used to collect data for a unique road segment in real life, and thus can be considered as a node, and the whole road network can be considered as multiple time-variant variables X with a graph G. Denote the whole time span of the datasets as T; then, $X\in R^{N\times T\times F}$, where N,F stand for spatial nodes and features, separately. Denote the history horizon of every sample as $T_{in}$, and predict the horizon as $T_{out}$. Then, sample sets can be generated from datasets X, which contain $T-T_{in}$ samples noted as $X=X_{t-T_{in}+1:t}$, where *t* stands for the latest time slice of each sample. The graph can be defined as G=(V,E,A), where *V* stands for spatial node sets, *E* stands for edge sets, and A stands for the adjacency matrix of the graph, where the values of the elements of *A* define both the existence and weight of the edges in *E*.

**Definition 1.** 
*Given multiple time-variant variables X with a graph G, the problem of $T_{out}$ steps forecasting using $T_{in}$ steps historical data of V time series can be defined as*

(6)
$${\hat{X}}_{t+1:t+T_{out}}=f\left(X_{t-T_{in}+1:t};G\right)$$



## 4. Dynamic Correlation Graph Convolutional Neural Networks

In this section, we introduce the DCGCN model proposed in this paper. First, we construct a correlation adjacency matrix using the statistic method mentioned in Section 3.1. Then, we put it in the graph convolution in the framework of the graph convolutional neural networks.

### 4.1. Method Overview

When using graph convolutional neural networks to deal with traffic prediction problems, a graph adjacency matrix is required to be input into graph convolution first. The core idea of this paper is also related to the construction of a graph adjacency matrix. Due to the lack of a graph adjacency matrix in some data sets or the limitations of the adjacency matrix itself, we extended the method of calculating the correlation coefficient in statistics to constructing the adjacency matrix, and make it related to the input sample to maintain its dynamics during construction. The above work is represented in the overall structure by “dynamic correlation matrices”. After obtaining the adjacency matrix, dynamic graph convolution and gated temporal convolution were carried out in the corresponding modules in the dynamic correlation layer. Finally, the results of each layer were processed through skip connection to complete the whole process.

### 4.2. Construction of Multiple Regression Dynamic Correlation Adjacency Matrix

The correlation matrix in multivariate time series can be formulated as
(7)$$r_{ij}=\frac{L_{x_{i}x_{j}}}{\sqrt{L_{x_{i}x_{i}}}\sqrt{L_{x_{j}x_{j}}}}=\frac{\sum_{t=1}^{T_{in}}\left(x_{it}-\bar{x_{i}}\right)\left(x_{jt}-\bar{x_{j}}\right)}{\sqrt{\sum_{t=1}^{T_{in}}{\left(x_{ik}-\bar{x_{i}}\right)}^{2}}\sqrt{\sum_{t=1}^{T_{in}}{\left(x_{jt}-\bar{x_{j}}\right)}^{2}}},$$
(8)$$R={\left(r_{ij}\right)}_{N\times N}=\left[\begin{array}{cccc} r_{11} & r_{12} & \cdots & r_{1N}\\ r_{21} & r_{22} & \cdots & r_{2N}\\ \vdots & \vdots & \cdots & \vdots \\ r_{N1} & r_{N2} & \cdots & r_{NN} \end{array}\right],$$
where *t* is the time slice in samples. Then, the correlation can be interpreted as similarity among different trends of time series of different nodes. We also used α as a threshold to control the sparsity.
(9)$$A_{DR}=\left\{\begin{array}{cc} r_{ij}, & r_{ij}\geq \alpha ,\\ 0, & otherwise, \end{array}\right.$$

Since it is generated using input data *X*, $A_{DR}\in R^{B\times N\times N}$ is sample-variant, which can be called a dynamic correlation adjacency matrix. The generated progress is in the dynamic correlation module (DCM) as shown in Figure 2.

### 4.3. Graph Convolutional Neural Networks for Traffic Forecasting

The whole model has an encoder–decoder structure. It has a dynamic correlation module to generate the adjacency matrices as shown in Figure 2, and a skip connection to prevent smoothing. The encoder and decoder are simple linear layers used to transform the shape of data. The hidden layers are several DCLayers, where each one contains a dynamic graph convolution module (DGCM, as shown in Figure 3), the difference between dynamic convolution and static convolution is shown in Figure 4. The DClayer also contains a gated temporal convolution module (GTCM, as shown in Figure 5), and the DClayer itself is shown in Figure 6. In addition, the whole structure is shown in Figure 7.

#### 4.3.1. Dynamic Graph Convolution Module

After generating the dynamic correlation adjacency matrix, it can be used for graph convolution to extract the spatial dependency between nodes. This kind of dynamic graph convolution can be defined as
(10)$$H^{\prime }=DGC\left(H,A\right)=A_{DR}HW+b,$$
where $A\in R^{B\times N\times N}$, $H\in R^{B\times N\times T\times C}$, and *B* is the batch size, which is the number of samples that are actually put in the model in a single iteration. *W* and *b* are learnable parameters. In comparison, as shown in Figure 4b, normal graph convolution (*with regard to not being dynamic*) can be defined as
(11)$$H^{\prime }=GC\left(H,A\right)=AHW+b,$$
where $A\in R^{N\times N}$, which means that it is the same in every sample in the iteration due to lacking the *B* dimension.

DGCM as shown in Figure 3 can be defined as
(12)$$H^{\prime }=DGCM\left(H,A\right)=A_{DR}HW_{1}+b_{1}+A_{A}HW_{2}+b_{2},$$
where $A_{A}$ is generated using the methods in Table 1.

#### 4.3.2. Gated Temporal Convolution Module

After using graph convolution to extract the spatial dependency, gated temporal convolution was used to extract temporal dependency as shown in Figure 5. It can be defined as
(13)$$H^{\prime \prime }=GTCM\left(H^{\prime }\right)=\sigma_{1}\left(H^{\prime }W_{1}+b_{1}\right)⊗\sigma_{2}\left(H^{\prime }W_{2}+b_{2}\right),$$
where $\sigma_{1}$, $\sigma_{2}$ activate functions tanh and sigmoid, ⊗ means the Hadamard product, and *W* and *b* are learnable parameters. During gated temporal convolution, parameter matrix $W_{1}$, $W_{2}\in R^{T_{l}\times T_{l+1}}$ are used to transform the amount of time slice $T_{l}$ in the current hidden layer to the next hidden layer $T_{l+1}$. Without padding, $T_{l+1}=T_{l}-2$ after 1-D convolution.

#### 4.3.3. Layer Model and Overall Structure

After a graph convolution through the spatial dimension and gated convolution through the temporal dimension, we added a residual connection to avoid over smoothing, constituting a DCLayer in hidden layers as shown in Figure 6, which can be defined as
(14)$$H_{l+1}=GTCM\left(DGCM\left(H_{l}\right)\right)+H_{l}^{\prime },$$
where $H_{l}^{\prime }\in R^{B\times N\times T_{l+1}\times C}$ is the middle $T_{l+1}$ time slices from $H_{l}\in R^{B\times N\times T_{l}\times C}$.

The overall structure of the model is the encoder–decoder structure. The encoder is an input layer that transforms the input feature *F* to hidden channel *C*, which can be defined as
(15)$$H_{1}=encoder\left(X\right)=XW+b.$$

After the encoder, *L* DCLayers were used to extract spatial and temporal dependencies. To acquire the result from different time scales to prevent over smoothing, a skip connection was added, which can be defined as
(16)$$H_{L}^{\prime }=\sum_{l=1}^{L}H_{l}^{\prime },$$
where $H_{l}^{\prime }\in R^{B\times N\times T_{L}\times C}$ is the middle $T_{l+1}$ time slices from $H_{l}\in R^{B\times N\times T_{l}\times C}$.

Finally, the decoder has two output layers, used to transform the time slice *T* and hidden channel *C*, which can be defined as
(17)$$Y=\hat{X}=decoder\left(H_{L}^{\prime }\right)=\left(H_{L}^{\prime }W_{1}+b_{1}\right)W_{2}+b_{2},$$
where $W_{1}\in R^{T_{L}\times 1}$, $b_{1}\in R$, $W_{2}\in R^{C{\times T}_{out}}$, $b_{2}\in R^{T_{out}}$ are learnable parameters.

Thus, the whole model as shown in Figure 7 can be defined as
(18)$$Y=\hat{X}=decoder\left({\left(DGCM\circ GTCM\right)}^{L}\left(encoder\left(X\right)\right)\right).$$

## 5. Experiment

We conducted an extensive experimental study to evaluate the effectiveness and efficiency of the proposed DCGCN models. The following sections are introduced in the following order. An introduction to the generated original data set is described in Section 5.1. In Section 5.2, experimental indicators, a comparison model, experimental results, and a comparative analysis are introduced in detail. All of these experiments were tested on Intel(R) Core(TM) i7-9750H CPU @ 2.60GHz and NVIDIA GeForce RTX 2060. The algorithms were implemented using Python 3.7 with PyTorch 1.7 and CUDA 10.1.

### 5.1. Generating Datasets

#### 5.1.1. Calculation of Features from Traffic Stream

The traffic stream has a certain pattern that is influenced by the roads’ status, space, and time. Typically, speed, flow, and density are used to describe traffic streams. These basic features can decrease the influence of roads, thus focusing more on space and time. The public datasets from PEMS [22] have all of these features, where the occupancy is similar to the density. The most common experiment from the traffic domain are speed prediction and flow prediction. Due to the easier and more accurate method of calculation, we chose flow as the feature of datasets.

The raw data of the original datasets are from an expressway networking toll data transmission system. The whole system produces millions of data per day, and each item has a large amount of specific information of vehicles, gantries, time stamps, etc. Thus, we cannot publish the datasets due to customers’ and geometric information. The calculation of traffic flow does not need specific data, but only the number of items, because flow is defined as the number of vehicles passing through a cross-section in the interval of time. The system has gantries to play the role of the cross-section, and the interval can be set to 5 min, which is the same as PEMS datasets. We named them HBD2 and HBD5.

The datasets of the experiment consisted of original datasets and public datasets. We introduce how we generated the original datasets in the next section. The statistics of the datasets are given in Table 2.

As shown in Table 2, we constructed the original dataset according to the standards of public data sets, and compared it in terms of several aspects: time slices, spatial vertices, features, the time span of the data source, included days, and the source. The number of time slices of the original data set is twice that of the public data set. This is because the data source of the original data set has a maximum time span of one month, but we make full use of it. They maintain the same order of magnitude in space vertices and the same feature.

#### 5.1.2. Calculation of Adjacency Matrix

Regarding the matrix or tensor of features, datasets used for graph convolution may also need an adjacency matrix. Although we do not have the distance on the graph to calculate the connectivity shown in Formula (6), we have the location of the gantries to calculate the distance between each pair of gantries using the haversine formula:(19)$$d_{ij}=2R\arcsin\left(\sqrt{\sin^{2}\left(\frac{\delta_{\theta }}{2}\right)+\cos\left(\theta_{i}\right)\cos\left(\theta_{j}\right)\sin^{2}\left(\frac{\delta_{\phi }}{2}\right)}\right),$$
where $\delta_{\theta }=\left|\delta_{i}-\theta_{j}\right|$ is the longitude difference, $\delta_{\phi }=\left|\phi_{i}-\phi_{j}\right|$ is the latitude difference, and *R* is the radius of the Earth, which is set to 6371 km. After calculating the distance matrix, we used Formula (Equation 19) to generate adjacency matrix A.

### 5.2. Experimental Studies

#### 5.2.1. Experimental Setting

Mean absolute error (*MAE*):(20)$$MAE\left(x,\hat{x}\right)=\frac{1}{\Omega }\sum_{i\in \Omega }|X_{i}-\hat{x_{i}}|,$$

Root mean square error (*RMSE*):(21)$$RMSE\left(x,\hat{x}\right)=\sqrt{\frac{1}{\Omega }\sum_{i\in \Omega }{\left(X_{i}-\hat{x_{i}}\right)}^{2}},$$
where Ω is the set that participates in the averages, and changes in different situations, such as |Ω|=N or |Ω|=N×T, which confuses the prediction of whether it is the average of all nodes from a single time slice or the average of nodes from multiple time slices. The experiments in this paper belong to the latter situation. Another commonly used evaluation matrix is the mean average percentage error (*MAPE*):(22)$$MAPE\left(x,\hat{x}\right)=\frac{1}{\Omega }\sum_{i\in \Omega }|\frac{X_{i}-\hat{x_{i}}}{x_{i}}|.$$

However, the feature of the datasets is the flow, which contains many zeros caused by rainy days, holidays, and other special circumstances (e.g., *COVID-19 lockdown*). Some of the codes of baseline models have not managed the dividing zero error, so the results only use the former two evaluation matrices.

#### 5.2.2. Baseline Models

Baseline models are listed by the published time as follows:DCRNN [22]: diffusion convolution recurrent neural network, which integrates graph convolution into an encoder–decoder gated recurrent unit.STGCN [23]: spatio-temporal graph convolutional network, which integrates graph convolution into a 1D convolution unit.ASTGCN [24]: attention-based spatial temporal graph convolutional network, which introduces spatial and temporal attention mechanisms into a model.GWN [25]: Graph WaveNet, which combines an adaptive adjacency matrix and 1D dilated convolution, which can handle long sequences.STSGCN [33]: spatial–temporal synchronous graph convolutional network, which uses localized spatial–temporal graphs to model localized correlations independently.MTGNN [26]: multivariate time series graph neural network, which uses graph learning, graph convolution, and temporal convolution modules to extract uni-directed relations in an end-to-end framework.AGCRN [27]: adaptive graph convolutional recurrent network, which uses node adaptive parameter learning module to capture node-specific patterns, and data-adaptive graph generation module to infer the inter-dependencies among different series.STFGNN [34]: spatial–temporal fusion graph neural network, which generates a “temporal graph” to compensate for correlations that spatial graphs may not reflect, and uses the fusion operation of various spatial and temporal graphs to learn hidden spatial–temporal dependencies.ST-Norm [36]: spatial and temporal normalization, which separately refine the high-frequency component and the local component underlying the raw data. Both modules can be integrated into other architectures.STGODE [37]: spatial–temporal graph ordinary differential equation network, which captures spatial–temporal dynamics through a tensor-based ordinary differential equation.

All of the results on the PEMS datasets of the above models come from papers in AAAI-21 and AAAI-22 [34,38].

#### 5.2.3. Main Results

The results of the original datasets, HBD2 and HBD5, are in Table 3. The results on public datasets, PEMS datasets, are in Table 4, where 15/30/60 means the number of minutes after the current time in the prediction, and the results in Table 4 are all predictions after one hour. We ran the codes that were provided by papers with original datasets; those with * mean that their model does not need a predefined adjacency matrix. GWN* is a special version of the ablation study in its paper. Although those models with * may have worse results, their requirements for datasets are lower, and the range of applications is expanded. Unless conditions such as AGCRN with HBD2 occur, the model is unavailable in this datasets because the results do not converge. In this case, the possible disadvantages of learning the graph structure completely through randomly initialized node embedding are reflected, which is not as stable as a predefined or calculated graph structure. Our model has two ways to choose from, so we can determine the approach based on the results. The STGCN and ST-Norm achieved better results on the HBD5 dataset than other datasets, comparing HBD2 with the public datasets in their respective papers, but this may be a special case. Their codes split the datasets by days, resulting in the need to drop part of the dataset to make the number of items divisible in order to run correctly. In addition, it can be divided into a training set, validation set, and test set with a correct ratio of 7:1:2. However, the economy and traffic of HBD5 may be easily affected by short-term external factors, or the excluded data may be be unstable data with large fluctuations. In short, this phenomenon makes this part of the results less credible, but should not affect the overall experimental results. The results on original datasets show that our model is better than baseline models, and the results on public datasets show that our model is 2% to 14% better than baseline models except for the AGCRN. Due to the condition mentioned above, the DCGCN has a wide application range, and is thus a better model. To summarize the results, we recognize that the DCGCN did not perform as well in the Table 3 original dataset as it did in the Table 4 public dataset with the best results from the DCGCN* or DCGCN in all datasets. However, the advantage of our model is that it has two choices. On the one hand, it can achieve the best accuracy on two original data sets, whereas the GWN, AGCRN, and ST-Norm can only achieve it on one data set. On the other hand, it can achieve good results even without an input of the adjacency matrix.

#### 5.2.4. Component Analysis

To verify the effectiveness of the components of the proposed model, we conducted a component analysis on the PEMS08 dataset to validate the effectiveness of key components. The models without different components are as follows:Without PA: the model without a predefined matrix uses a dynamic correlation matrix in the DGCM.Without DC: the model without a dynamic correlation matrix uses a predefined matrix instead in the DGCM.Without GC: the model without graph convolution uses a linear layer instead.Without TC: the model without gated temporal convolution uses a linear layer instead.Without SP: the model without special connections, such as residual connection.Final: the final model with all components.

The roles of each components in DCLayer are shown in Figure 8, and the results in the component analysis of each model are shown in Table 5. We can draw three conclusions: (1) the model without PA performs better than the model without DC, which means that the dynamic correlation matrix can play the role of a predefined matrix when the latter is unavailable under certain circumstances without affecting the performance of the model; (2) the models without GC or TC perform far worse than other models because these two are key components in most GNN models in the traffic domain. Furthermore, we can see from the comparison of the results of the train loss, valid loss and test MAE that the former model is under-smooth and the latter is over-smooth; (3) the train loss and valid loss of the model without SP are similar to the final model, but the test MAE is approximately 17% higher, which means that the model is severely over-smooth. This verifies the effectiveness of special connections to prevent over-smoothing.

More details of the results of the above models are shown in Figure 9. (1) The curves of the model without PA are similar to the final model, which shows that the DCGCN can achieve a certain effect without a predefined adjacency matrix. In comparison, the model without DC only use a predefined matrix, which limits the spatial dependencies to have inferior results. (2) Both models without GC or TC cannot have good results, but their curves are very different due to the different circumstances of under-smoothing and over-smoothing. The former’s curves are gentle and slow to converge whereas the latter’s curves are fast to converge in training loss and validation loss but unstable in test MAE. (3) The model without SP is severely over-smooth; thus, its curve in test MAE becomes unstable after only 30 epochs, which shows that special connections are crucial as well.

## 6. Conclusions

In this paper, we studied the multivariate time series prediction problem and present a novel framework for spatial–temporal traffic data forecasting. The core idea of the model is to construct different similarity adjacency matrices according to different dynamic samples to carry out graph convolution, so as to capture the dynamics of space variably. Our model can deal with the dataset regardless of whether it has a predefined graph structure or not. By combining a predefined adjacency matrix, dynamic correlation matrix, and adaptive matrix, the DCGCN can learn global and localized spatial–temporal dependencies through spatial graph convolution and temporal convolution. The extensive tests on two original datasets and four public datasets verified the superiority of the proposed solutions in this paper. In the future, we can try to extend this method to the time dimension to further capture the dynamics of space–time in all aspects.

## Figures and Tables

**Figure 1 sensors-23-02897-f001:**
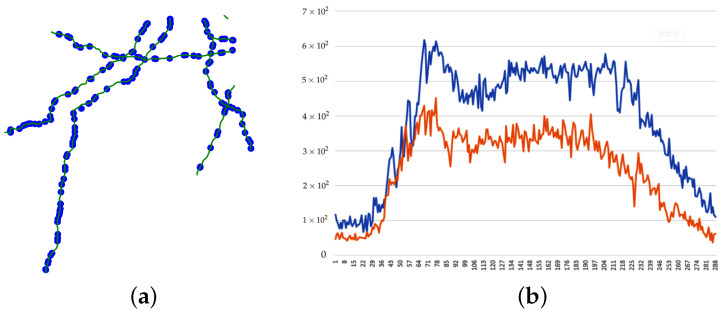
Traffic data. (**a**) Road network. (**b**) Two nodes’ features in one day.

**Figure 2 sensors-23-02897-f002:**
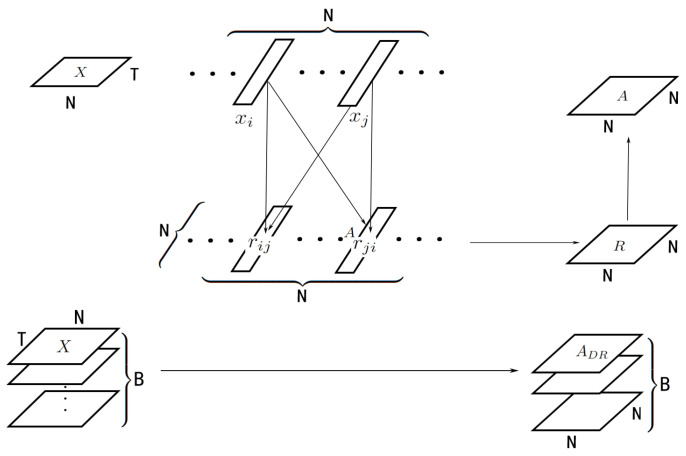
DCM divides input *X* into *N* parts, where every pair of parts interact with each other, formulated as a square matrix. Different input *X* turns into different $A_{DR}$.

**Figure 3 sensors-23-02897-f003:**
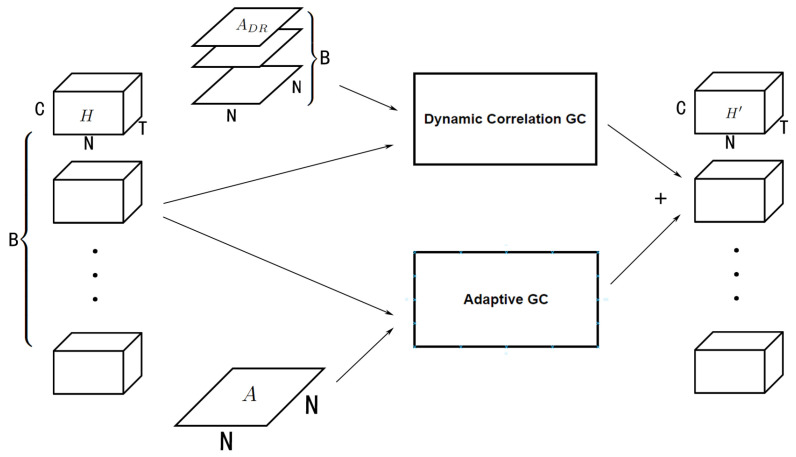
DGCM has a dynamic correlation graph convolution and an adaptive graph convolution that use adjacency matrices $A_{DR}$ that DCM generated and adaptive adjacency matrix A, respectively. DGCM combines the result of the two operations so that the shape of the input and output stays the same.

**Figure 4 sensors-23-02897-f004:**
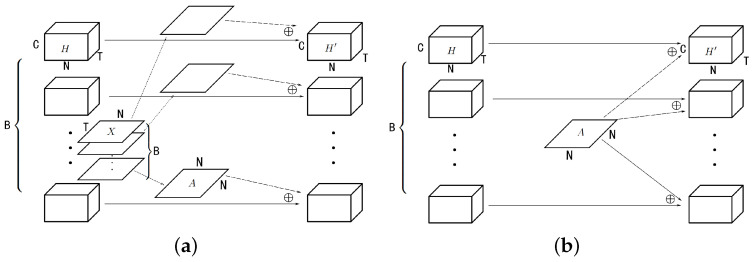
Dynamic and static convolution. Dynamic convolution uses a different adjacency matrix *A* that is generated from different input *X*, whereas static convolution uses the same *A* that is predefined or trained. ⊕ stands for the Einstein summation convention used in the multi-dimensional operation. (**a**) Dynamic. (**b**) Static.

**Figure 5 sensors-23-02897-f005:**
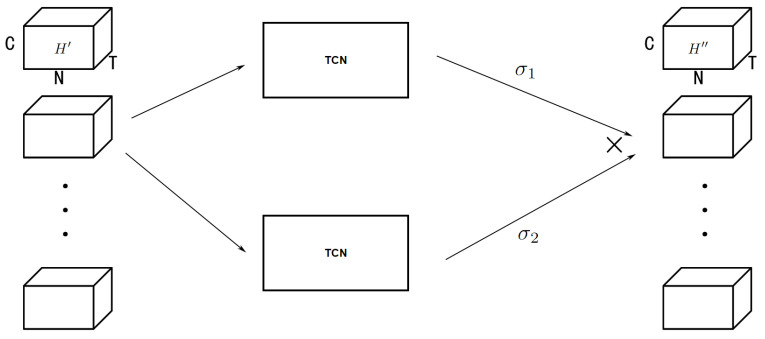
GTCM.

**Figure 6 sensors-23-02897-f006:**
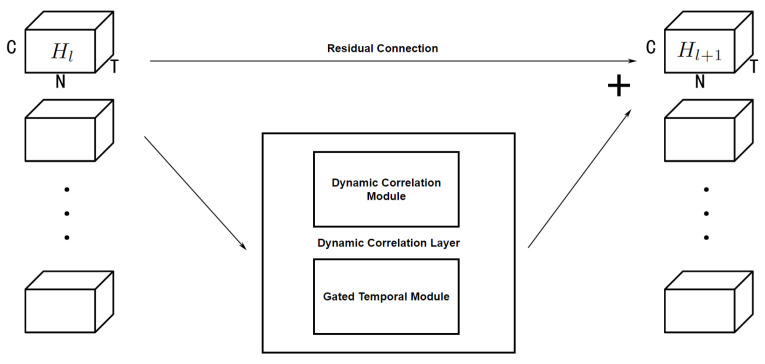
DCLayer.

**Figure 7 sensors-23-02897-f007:**
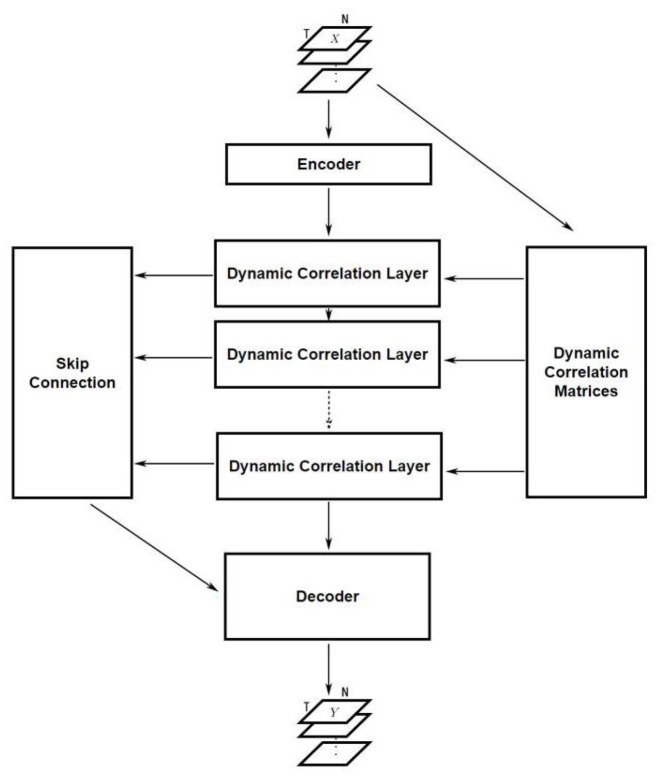
Overall structure.

**Figure 8 sensors-23-02897-f008:**
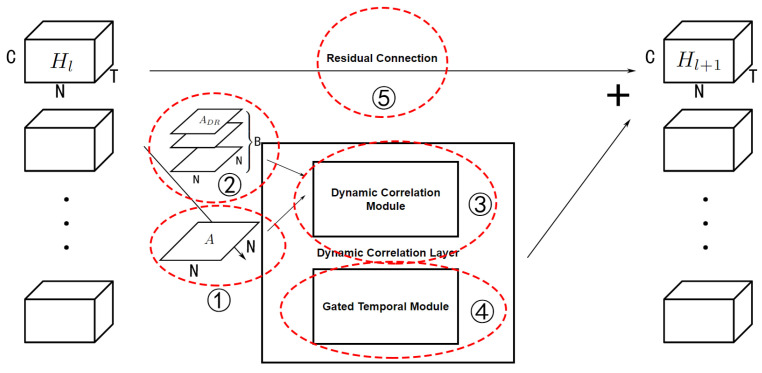
The role of each components in DCLayer.

**Figure 9 sensors-23-02897-f009:**
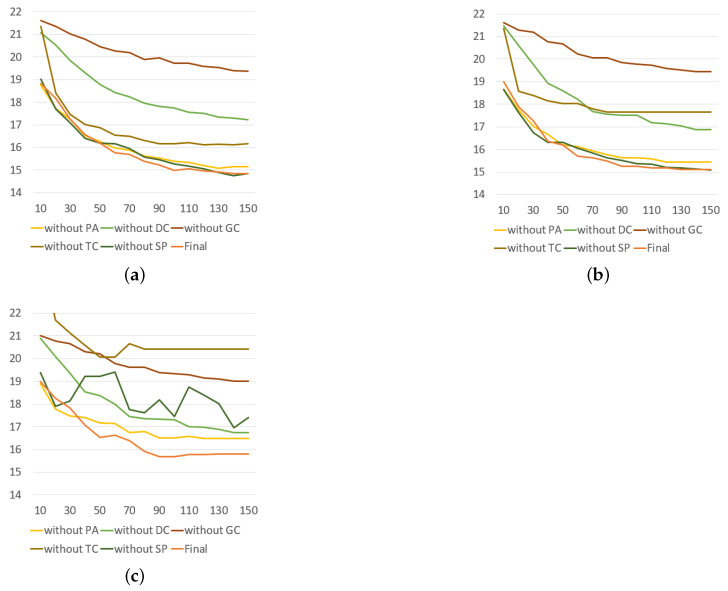
Rate of convergence in component analysis. (**a**) Training loss. (**b**) Validation loss. (**c**) Test MAE.

**Table 1 sensors-23-02897-t001:** Methods for constructing adjacency matrix.

Model	Method	Equation
-	Global Adjacency Matrix	$A=\sigma \left(W\right)$
AGCRN	Undirected Adjacency Matrix	$A=\sigma \left(\alpha \left(W_{1}W_{1}^{T}\right)\right)$
GWN	Directed Adjacency Matrix	$A=\sigma \left(\alpha \left(W_{1}W_{2}^{T}\right)\right)$
MTGNN	Uni-directed Adjacency Matrix	$A=\sigma \left(\alpha \left(W_{1}W_{2}^{T}-W_{2}W_{1}^{T}\right)\right)$
SLCNN	Structure Learning Adjacency Matrix	$A=\sigma (\left(X_{t}W_{1}\right)\left(W_{2}^{T}X_{t}^{T}\right))$
MTGNN	Dynamic Adjacency Matrix	$A=\sigma \left(X_{t}W_{1}X_{t}^{T}\right)$

**Table 2 sensors-23-02897-t002:** Statistics of datasets.

Dataset	Time Slices	Space Vertices	Feature	Time Span	Number of Days	Source
HBD2	8928	159	Volume	1 January 2021–31 January 2021	31	Original
HBD5	8928	213	Volume	1 January 2021–31 January 2021	31	Original
PEMS03	26208	358	Volume	1 September 2018–30 November 2018	91	STSGCN
PEMS04	16992	307	Volume, Density, Speed	1 January 2018–28 February 2018	59	ASTGCN
PEMS07	28224	883	Volume	1 May 2017–31 August 2017	approximately 123	STSGCN
PEMS08	17856	170	Volume, Density, Speed	7 July 2016–31 August 2016	62	ASTGCN

**Table 3 sensors-23-02897-t003:** Experiment results on original datasets.

Model	HBD2 (159)						HBD5 (213)					
	15		30		60		15		30		60	
	MAE	RMSE	MAE	RMSE	MAE	RMSE	MAE	RMSE	MAE	RMSE	MAE	RMSE
STGCN	4.9870	7.1400	5.2850	7.9950	6.1800	10.2320	4.3160	6.6310	4.6250	7.2320	5.3260	8.4350
GWN *	4.7039	6.9059	4.8391	7.2365	5.0874	7.8387	4.4142	6.5950	4.5325	6.8892	4.7487	7.3685
GWN	4.6813	6.8511	4.8089	7.1535	5.0429	7.6974	4.4265	6.5593	4.5687	6.8731	4.7967	7.3375
MTGNN *	4.8073	7.0080	4.9065	7.2461	5.1076	7.7167	4.4166	6.5704	4.5198	6.7981	4.6964	7.1670
AGCRN *	6.3733	11.5733	6.5017	11.8567	6.6717	12.1825	4.3233	6.6133	4.3833	6.7717	4.5167	7.0683
STSGCN	5.8982	8.7210	5.9318	8.7839	6.0038	8.9148	6.0371	8.9229	6.0644	8.9839	6.1335	9.1175
STFGNN	4.9325	7.2288	4.9877	7.3545	5.0805	7.5689	5.3253	8.1867	5.3429	8.2264	5.3857	8.3056
ST-Norm *	4.8563	6.9447	4.9343	7.1255	5.1332	7.5475	4.3227	6.4130	4.4518	6.6590	4.7009	7.0890
DCGCN *	4.7465	6.9822	4.8633	7.2755	5.0765	7.7819	4.3930	6.5613	4.5147	6.8348	4.7323	7.2662
DCGCN	4.6937	6.9157	4.8217	7.2329	5.0547	7.7841	4.3619	6.4932	4.4890	6.7795	4.6909	7.1848

Those models with * mean that they don’t need a predefined adjacency matrix.

**Table 4 sensors-23-02897-t004:** Experiment results on public datasets.

Model	PEMS03		PEMS04		PEMS07		PEMS08	
	MAE	RMSE	MAE	RMSE	MAE	RMSE	MAE	RMSE
DCRNN	18.18	30.31	24.70	38.12	25.30	38.58	17.86	27.83
STGCN	17.49	30.12	22.70	35.55	25.38	38.78	18.02	27.83
ASTGCN	17.69	29.66	22.93	35.22	28.05	42.57	18.61	28.16
GWN	19.85	32.94	25.45	39.70	26.85	42.78	19.13	31.05
STSGCN	17.48	29.21	21.19	33.65	24.26	39.03	17.13	26.80
LSGCN	17.94	29.85	21.53	33.86	27.31	41.46	17.73	26.76
STFGNN	16.77	28.34	19.83	31.88	22.07	35.80	16.64	26.22
STGODE	16.50	27.84	20.84	32.82	22.59	37.54	16.81	25.97
AGCRN	15.98	28.25	19.83	32.26	22.37	36.55	15.95	25.22
DCGCN *	15.41	26.03	**19.81**	**31.11**	24.51	37.75	16.49	25.46
DCGCN	**15.29**	**25.98**	20.28	31.65	**22.06**	**34.66**	**15.68**	**24.39**

Those models with * mean that they don’t need a predefined adjacency matrix.

**Table 5 sensors-23-02897-t005:** Results of component analysis.

Model	Training Loss	Validation Loss	Test MAE
without PA	15.1482	15.4286	16.4865
without DC	17.2296	16.885	16.7511
without GC	19.3766	19.4391	19
without TC	16.1691	17.6542	20.4214
without SP	14.8499	15.0907	17.4012
Final	15.0424	15.2778	15.9937

## Data Availability

Due to the nature of this research, participants of this study did not agree for their data to be shared publicly, so supporting data are not available.

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
