# Peer review of "Dynamic Correlation Adjacency-Matrix-Based Graph Neural Networks for Traffic Flow Prediction"

_sensors, 2023, doi:10.3390/s23062897_

Round 1

Reviewer 1 Report

Dear Authors,

Your paper entitled Dynamic Correlation Adjacency Matrix Based Graph Neural Networks for Traffic Flow Prediction bring a new way for traffic flow prediction. It really worthwhile to understand traffic flow nowadays. Predictions can help a lot in the future and I appreciate your efforts in this direction.

The paper has scientific soundness for sure, the math is clean and the construction Construction of Multiple Regression Dynamic Correlation Adjacency Matrix is elegant. Figure 4 really helps visualize differences between static and dynamic convolutions.

From the perspective of one researcher to another the purpose the paper is fine. However, you have to keep in mind that you audience of you research is larger. Thus said, I recommend that you take a step back and overlook chapter 5. 

From the perspective of a new reader, outside of this domain or new to the domain, the paper seem dry and your results were not valued enough, which should not be the case in your paper. Table 2 show evaluation of 6 datasets. Insert some paragraphs to conduct the reader to your conclusions. Send your reader to further reading if time is too short to make changes to the manuscript.

Other recommendations are the following:

1. Figure 9 can be enlarged or split in order to better understand all the options (components) that your  Dynamic Correlation Graph Convolutional Neural Network can lead to different results. Further analysis of the results and discussions regarding a specific dataset and these options will help you also understand why are the so much differences between two datasets.

2. Some structure in chapter 5 will bring readability especially where you present your results.

3. Chapter 6 is too short and has to leave the reader with definite answers in order to fulfill the purpose of making your research known.

Best of luck with your research! I hope you reach to the same conclusion.

Author Response

Comment 1.Chapter 5 need to be overlooked that from the perspective of a new reader, outside of this domain or new to the domain, the paper seem dry and your results were not valued enough, which should not be the case in your paper. Table 2 show evaluation of 6 datasets. Insert some paragraphs to conduct the reader to your conclusions.

Response: Thanks for your suggestions. We have included the following description below Table 2 to help new readers better understand the data sets in the table. We have revised all the minor errors existing in our table that may cause confusion.

As shown in Table 2, we constructed the original dataset according to the standards of public data sets, and compared it in several aspects: time slices, spatial vertices, feature, time span of the data source, included days, and source. The number of time slices of the original data set is twice that of the public data set. This is because the data source of the original data set has a maximum time span of one month, but we make full use of it. They maintain the same order of magnitude in space and the same characteristics.

Comment 2. Figure 9 can be enlarged or split in order to better understand all the options (components) that your Dynamic Correlation Graph Convolutional Neural Network can lead to different results. Further analysis of the results and discussions regarding a specific dataset and these options will help you also understand why are the so much differences between two datasets.

Response:   Many thanks for your valuable comment. We have enlarged the arrangement of the three images in Figure 9 from one line to two lines to better allow the reader to observe the differences. Besides, we have changed the layout to reduce the white space in the article.

Comment 3. Some structure in chapter 5 will bring readability especially where you present your result

Response:   Thanks for your advice. In our revised version, we added the description of the structure of chapter 5 to improve the readability of the resulting section.

The following sections are introduced in the following order. An introduction to the generated original data set is described in 5.1. In 5.2, experimental indicators, comparison model, experimental results and comparative analysis are introduced in detail.

Comment 4. Chapter 6 is too short and has to leave the reader with definite answers in order to fulfill the purpose of making your research known.

Response:  Thanks for your suggestion. In our revised manuscript, we have added a description of the core idea of the paper and the prospect of future research directions in Chapter 6 to support the whole paper.

“The core idea of the model is to construct different similarity adjacency matrices according to different dynamic samples to carry out graph convolution, so as to capture the dynamics of space variably.” … “ In the future, we can try to extend this method to the time dimension to further capture the dynamics of space-time in all aspects.”

Reviewer 2 Report

1.The structure of the article writing is appropriate, the experimental design is complete and the experimental results can support the inference of the method.

2.The description part of the method proposed in the paper is suggested to be improved. The current writing method cannot allow readers to clearly correspond from the overall structure to the method of each sub-step.

3.The experimental results currently only compare the MSE and RMSE of various methods. Can you provide other indicators to prove the effectiveness of the proposed method?

4.From the experimental results, it is observed that DCGCN cannot obtain the best effect in all test data sets or the effect improvement is not very significant. Have the authors considered adding experimental comparisons with more datasets? Or provide other evidence to prove the excellence of the method proposed in this paper?

Author Response

Comment 1. .The structure of the article writing is appropriate, the experimental design is complete and the experimental results can support the inference of the method.

Response: Many thanks for your comments. We spent a lot of energy on the experimental designing and evaluating our methods. It’s our honor to receive your approval.

Comment 2.  The description part of the method proposed in the paper is suggested to be improved. The current writing method cannot allow readers to clearly correspond from the overall structure to the method of each sub-step.

Response: Thank you very much for your advice. We realized that the description in Chapter 4 did not give the reader a picture of the whole before describing it in detail, so we added the method overview at the beginning of Chapter 4, connecting the small modules to the overall structure.

“When using graph convolutional neural networks to deal with traffic prediction problems, graph adjacency matrix is required to input into graph convolution first. The core idea of this paper is also related to the construction of graph adjacency matrix. Due to the lack of graph adjacency matrix in some data sets or the limitations of the adjacency matrix itself, we extend the method of calculating correlation coefficient in statistics to construct the adjacency matrix, and make it related to the input sample to maintain its dynamics during construction. The above work is represented in the overall structure by "Dynamic Correlation Matrices". After obtaining the adjacency matrix, Dynamic Graph Convolution and Gated Temporal Convolution are carried out in the corresponding modules in the Dynamic Correlation Layer. Finally, the results of each layer are processed through skip connection to complete the whole process.”

Comment 3. The experimental results currently only compare the MSE and RMSE of various methods. Can you provide other indicators to prove the effectiveness of the proposed method?

Response:  Many thanks for your comments. We are sorry that we only provide two evaluation criteria and in some related articles they also provide MAPE. That's because we ran into some of the same problems in calculating MAPE that other research teams have run into with only two criteria. Behind the verification of multiple comparison models are the public code that needs to be run for the paper. Different code uses different Python frameworks, such as TensorFlow, pytorch, and mxnet, and even for the same framework, different articles use different versions depending on the publication year. This can lead to an anomaly in the calculation of MAPE. Even though MAPE is mathematically determined, the way it implements the program is different. For example, the TensorFlow 1.X version used by STGCN does not deal with the division by 0 exception, which makes it impossible to compare the results of other models when working with data sets containing 0. The technical reasons for this are not described in detail in the paper, but they are only part of the reason. On the other hand, as we wrote in the introduction to MAPE in 5.2.1, since the original data set was generated by the team processing the original charge data, there was no official website and system support like PEMS, and some external environmental factors affected the data set, so MAPE was not applicable to the experimental results. I hope the above explanations may help you better understand our work.

Comment 4. From the experimental results, it is observed that DCGCN cannot obtain the best effect in all test data sets or the effect improvement is not very significant. Have the authors considered adding experimental comparisons with more datasets? Or provide other evidence to prove the excellence of the method proposed in this paper?

Response: Thank you very much for your comments. We recognize that DCGCN did not perform as well in Table 3 original dataset as it did in the Table 4 public dataset with the best results from DCGCN* or DCGCN in all datasets. However, as discussed in 5.2.3, the advantage of our model is that it has two choices. On the one hand, it can achieve the best accuracy on two original data sets, while GWN, AGCRN and ST-Norm can only achieve it on one data set. On the other hand, it can achieve good results even without input of adjacency matrix. Therefore, we believe that the existing data can support DCGCN, which is a better model. The above description will be included in 5.2.3 as a summary. Meanwhile, we will also focus on more experiments in the future to obtain more evidence.

“To summarize the results, we recognize that DCGCN did not perform as well in the Table 3 original dataset as it did in the Table 4 public dataset with the best results from DCGCN* or DCGCN in all datasets. However, the advantage of our model is that it has two choices. On the one hand, it can achieve the best accuracy on two original data sets, while GWN, AGCRN and ST-Norm can only achieve it on one data set. On the other hand, it can achieve good results even without input of adjacency matrix.”

Reviewer 3 Report

Authors applied the graph neural networks for traffic flow prediction. Six datasets are used to prove the favourable performance of new model. And the paper is well organized. So i recommend it for publication in sensors.

Author Response

Comment 1. Authors applied the graph neural networks for traffic flow prediction. Six datasets are used to prove the favourable performance of new model. And the paper is well organized. So i recommend it for publication in sensors.

Response: Thank you very much for your comments, we are very honored to receive your recognition, we really put a lot of time and effort into the conception of the model, building the original data set and various trials, we are very happy to see these efforts recognized. Finally, I would like to thank you again and wish you a better career in scientific research.

Round 2

Reviewer 2 Report

no comments